# Ice Volumes in Permafrost Landscapes of Arctic Yakutia

**Alexander N. Fedorov** [1,*] , **Pavel Y. Konstantinov** [1,2], **Nikolay F. Vasiliev** [1,2], **Nikolay I. Basharin** [1,2] ,
**Andrei G. Shepelev** [1,2] , **Varvara A. Andreeva** [1,2], **Valerii P. Semenov** [1,2], **Yaroslav I. Torgovkin** [1,2],
**Alexey R. Desyatkin** [1,3] , **Mikhail N. Zheleznyak** [1] and **Igor P. Semiletov** [2,4]

1    Melnikov Permafrost Institute of the Siberian Branch of the RAS, 677010 Yakutsk, Russia
2    Laboratory for Integrated Research of the Arctic System "Land-Shelf", Tomsk State University,
      634050 Tomsk, Russia
3    Institute for Biological Problems of Cryolithozone SB RAS, 677980 Yakutsk, Russia
4    V.I. Il'ichov Pacific Oceanological Institute, Far Eastern Branch of the Russian Academy of Sciences,
      690041 Vladivostok, Russia
*    Correspondence: fedorov@mpi.ysn.ru; Tel.: +7-914-224-9032

**Abstract:** This article is devoted to the study of the distribution of ground ice volumes in the upper layers of 5–10 m permafrost in the permafrost landscapes of Arctic Yakutia. Compilation of such a map will serve as a basis for assessing the vulnerability of permafrost to global warming, anthropogenic impact and forecasting the evolution of permafrost landscapes. The map was compiled using ArcGIS software, which supports attribute table mapping. The ground ice map of Arctic Yakutian permafrost landscapes shows that about 19% of the area is occupied by ultra ice-rich (above 0.6 in volumetric ice content) sediments. Very high ice volumes (0.4–0.6) are cover approximately 27%, moderate ice volumes (0.2–0.4)—25% of the area, and low ice volumes (less than 0.2)—about 29% of Arctic Yakutia.

**Keywords:** ground ice; ice content; mapping; permafrost landscape; Arctic Yakutia

## 1. Introduction

Current global warming is most pronounced in the Arctic and Subarctic regions. In northern Yakutia located within the Arctic and Subarctic zone of Central and North-Eastern Siberia, the mean annual air temperature in 2020 was, on average, 3 °C higher compared to 1960–1970. For the Russian Arctic, a 7 °C increase in air temperature is predicted by the end of the 21st century relative the late 20th century [1]. This warming has a strong impact on permafrost which is the primary factor shaping the landscape sphere. The degradation or aggradation of permafrost have important implications for all landscape processes. The geoecological state of the Arctic environment depends entirely on the degree of landscape transformation by permafrost processes. The response of the natural environment to warming is observed both in the thermophysical state [2–5] and landscape morphology [6–8]. Anthropogenic impact in a warming climate can exacerbate environmental and social problems [9–12].

The Arctic landscapes are particularly vulnerable to climate change and human impacts because of the presence of ground ice in underlying permafrost. The melting of ground ice can lead to rapid changes in the content and morphology of Yakutian permafrost landscapes [13–17]. Permafrost processes have been adopted by the Russian Ministry of Economic Development (Guidelines no. 267 of 13 May 2021) as criteria for assessing climatic change risks. The rates, extent, duration, and effect of these processes depend on the type and volume of ground ice. Ice content of sediments is an important characteristic in estimating the transformation of permafrost landscapes. In the 1970s and early 1980s, ground ice content was recognized as the main criterion for sensitivity of the surface (or permafrost landscapes) to man-made impacts [18–24].

Knowledge of the geographical distribution of massive ground ice bodies (ice wedges, tabular ice, and buried ice) and segregated (cryostructural) ground ice, as well as their

volumes and relation to permafrost processes is important to understanding the evolution of permafrost environments associated with global warming and anthropogenic impacts. This determines the significance of geocryology among other disciplines and its importance for the socio-economic system in permafrost regions. Determining the adaptation capacity of social life, infrastructure and economic systems to the warming climate in permafrost regions is one of the most important issues in the modern world.

Information on cryostructures, wedge ice and volumetric ice contents is presented in the permafrost maps for the USSR (Russia) published previously, including the Cryolithological Map of the USSR, 1:4,000,000 scale [25] and the Geocryological Map of the CIS, 1:2,500,000 scale [26]. The Permafrost-Landscape Maps of Yakutia [27,28] also contain information on ground ice conditions. Smaller-scale maps are available for various regions of Yakutia showing the cryostratigraphic characteristics of permafrost [24,29–31].

However, new data on the ground ice conditions have been collected and mapping technologies have advanced since then. Predicted environmental changes due to global climatic warming require modification of the purpose and objectives of ground ice mapping. The main purpose of this study is to provide information on ground ice volumes in the upper 5–10 m of permafrost for subsequent interpretation and prediction of the evolution of permafrost landscapes in Arctic Yakutia in response to global climatic warming.

## 2. Materials and Methods

The study region covers part of the Republic of Sakha (Yakutia) north of the Arctic Circle, encompassing about 1.3 million km². Administratively, it comprises ten municipalities of Yakutia included into the Arctic Economic Zone of the Russian Federation. The landscape of Arctic Yakutia consists of tundra and boreal lowlands, the Verhkoyansk and Chersky mountain systems, and high uplands of the Central Siberian Plateau and North-Eastern Siberia.

To map the ground ice conditions in permafrost landscapes of Arctic Yakutia, we used the previously published permafrost landscape maps of the Republic of Sakha (Yakutia) [27,28] as an information base. The main method for synthesizing permafrost data was to integrate two landscape-forming factors, geological-geomorphological and biohydroclimatic, by combining two thematic layers, terrain type and vegetation group, in the ArcGIS platform. Ground temperature, ice content, active layer thickness and permafrost processes are closely interrelated in any given land unit and their combination as a system of components forms a distinctive permafrost landscape [32].

A terrain type is defined as a part of the land formed by a particular genetic (or stratigraphic-genetic) type of sediments (alluvial, lacustrine-alluvial, fluvioglacial, residual, etc.), or part or combination thereof, which has characteristic topographic setting and microsite combination. For Yakutia, we identified and mapped 20 terrain types shown at a 1:1,500,000-scale map of permafrost landscapes [28].

A vegetation group is a bioclimatic complex, uniform in respect to morphological structure and physico-geographical processes, which has characteristic heat and water relationships and distinct types of soil and vegetation. In total, we have identified 35 vegetation groups, including arctic graminoid tundra, typical shrub and moss-lichen tundra, southern shrub tundra, patterned fen, larch shrub-moss-lichen open woodland, boreal scarce larch bogs, etc.

Each terrain–vegetation combination has characteristic permafrost and landscape parameters. For this mapping effort, we used the inventory of permafrost landscapes compiled earlier [33] to integrate the geocryological database with the GIS attribute table. Input data for each permafrost parameter in the attribute table were grouped and summarized.

In compiling the permafrost landscape maps of Yakutia [27,28], ground ice volumes were calculated as a sum of volumetric ice content of permafrost block samples and a volume of wedge ice, if available. Ice contents of permafrost blocks were determined based on the dry unit weight of a frozen soil and total gravimetric moisture content [34,35].

Wedge ice volumes were calculated using the approach developed by Gasanov [36] with consideration for polygon size.

Data on ground ice volumes for the compilation of the above maps were drawn both from published [31,37–46] and unpublished sources available at the Melnikov Permafrost Institute. The unpublished sources included the reports of the First Northern Expedition led by N.F. Grigoriev (1954–1957) and the Second Northern Expedition led by Y.V. Shumilov (1984–1987), as well as field records by E.M. Katasonov, I.A. Nekrasov, P.P. Gavriliev, I.V. Klimovsky, V.V. Kunitsky and many other researchers.

The ground ice data for Arctic Yakutia were classified on the basis of stratigraphic-genetic types of surficial deposits in the landscape zones using the major permafrost summary works [25–27,33,47,48]. In mapping permafrost ice conditions, four classes are commonly used for ground ice content: ice-poor (20% ice, or 0.2), ice-rich (20–40%, or 0.2–0.4), very ice-rich (40–60%, or 0.4–0.6) and ultra ice-rich (over 60%, or 0.6) permafrost [49]. The data were constructed into an ArcGIS system and the attribute table was used to compile a map of ice contents in surficial material.

The Permafrost-Landscape Map of the Republic of Sakha (Yakutia) at scale 1: 1,500,000 created on the ArcGIS platform was used as the base map for the compilation (https://cloud.mail.ru/public/8j62/vJpw5hEGa/mlk_2020.pdf accessed on 7 December 2022). In all, 5720 polygons were identified in Arctic Yakutia which were grouped into four ice content categories. There are some uncertainties in the map related to permafrost dynamics. For example, the map does not show areas of degrading ice-rich permafrost landscapes. Further work needs to be done to resolve the task.

### 3. Results

In Arctic Yakutia, 19 terrain types were identified on the basis of genetic (or stratigraphic–genetic) complexes of sediments controlling the differentiation of permafrost landscapes by dominant cryostructure and volumetric ice content of surficial material (Figure 1, Table 1). The terrain types consist of landforms ranging from lowlands to alpine-type features, surficial materials ranging from peat to clastic and solid rocks, and vegetation ranging from bogs to mountain deserts [28].

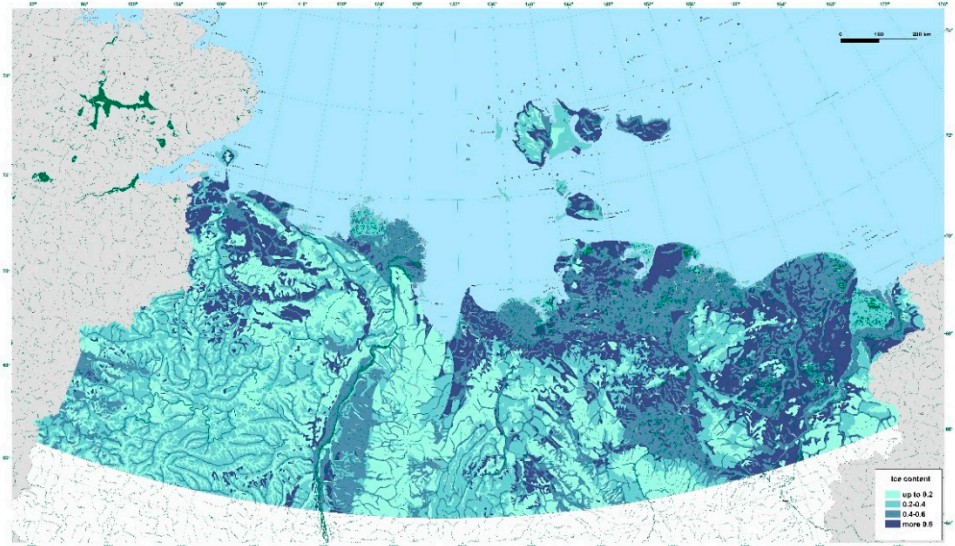

**Figure 1.** Ground ice content map of Arctic Yakutia (https://cloud.mail.ru/public/8j62/vJpw5hEGa/Ground_ice_content_of_Arctic_Yakutia.pdf accessed on 7 December 2022).

**Table 1.** Prevailing cryostructure and Volumetric ice content in terrain types, Arctic Yakutia.

| | Terrain Type | Stratigraphic-Genetic Complex | Prevailing Cryogenic Textures and Trapped Ice | Volumetric Ice Content |
|---|---|---|---|---|
| 1 | Marshes | mH, mH1-2 | Massive, lenticular | 0.2–0.4 |
| 2 | Low terrace | aH, aIII-H | Massive, lenticular, layered; Holocene ice wedges | 0.4–0.6 |
| 3 | Mid terrace | aII-III | Massive; occasional tabular ice | 0.2–0.4 |
| 4 | Inter-ridge-lowland | aII-III, bH | Massive, lenticular, layered | 0.4–0.6 |
| 5 | High terrace | aI, aI-II | Massive | up to 0.2 |
| 6 | Inter-alas | LedIII, laII-III, aIII | Layered, lenticular, reticular; Pleistocene ice wedges | more 0.6 |
| 7 | Alas | lpH, LH | Layered, lenticular, reticular; Holocene ice wedges | 0.4–0.6 |
| 8 | Moraine | gIII | Massive, cortical, basal; polygonal ice wedge systems | 0.4–0.6 |
| 9 | Outwash | fIII | Massive, lenticular, cortical | 0.2–0.4 |
| 10 | Eluvial and Rock-ridge * | e, ed and r | Lenticular, massive, layered, cortical and crack-like ice | up to 0.2 |
| 11 | Colluvial | c | Massive, cortical, basal | up to 0.2 |
| 12 | Deluvial colluvial | dc | Massive, cortical, lenticular | up to 0.2 |
| 13 | Deluvial solifluction | ds, dp | Layered, lenticular, massive, reticular | 0.2–0.4 |
| 14 | Non-drained mari | bH, e, ed | Layered, lenticular, reticular; Holocene ice wedges | more 0.6 |
| 15 | Glacial-valley | gIII | Massive, cortical, basal | 0.2–0.4 |
| 16 | Glacier | - | Glacier ice | more 0.6 |
| 17 | Tukulan (eolian) | v | Massive | up to 0.2 |
| 18 | Valley-sea terrace | amIII | Massive, lenticular; occasional tabular ice | 0.2–0.4 |

* Eluvial and Rock-ridge terrain types are merged unlike the previous Permafrost landscape map of Yakutia [28].

GIS modeling using the attribute table made it possible to systematize the ice content of surface materials both in terms of ice content and geographical space (Table 2).

Permafrost landscapes with high ice contents in excess of 0.4 occupy about 46% of Arctic Yakutia.

About 19% of the map region consists of ultra ice-rich (>0.6) landscapes. The inter-alas terrain type with an ice complex is prominent here, represented by yedoma in lowlands. Geographically, the ice complex occurs in the Anabar–Olenek interfluve, the New Siberian Islands and the Primorsky lowland, as well as in the basins of the Yana, Indigirka and Kolyma rivers. Wedge ice can comprise up to 63% of the ice-complex volume [50]. The distribution of the ice complex is well documented by Strauss [51–53]. The cryolithology of the ice complex with thick ice wedges has long been the focus of scientific interest [37,38]. Well-known classical exposures include Duvanny Yar on the Kolyma Riverbank [54], Oyogos Yar on the Laptev Strait [55], Cape Mamontov Klyk [56], Moustakh Island in the Tiksi Bay [57], Batagaika [58,59] and others. The ice complex can be diverse in origin [60].

Non-drained mari are also classified as ultra ice-rich landscapes. Classical examples can be found in poorly drained flat watersheds between the Olenek and Lena Rivers on the Central Siberian Plateau, such as the so-called Eyk Formation consisting of mantle clay and mud with interbeds of plant detritus, peat, erratic pebble and containing ice wedges. The peatland is as thick as 4 m in places. The ice content of sediments is up to 70–90% by

volume [33]. Glaciers occur in the Orulgan Ridge [61], along the main water divide. There are 80 glaciers in total area of 20 km$^2$.

**Table 2.** Volumetric ice content in terrain types, Arctic Yakutia.

| Volumetric Ice Content | Terrain Type | Area, km$^2$ | % | Area, km$^2$ | % |
|---|---|---|---|---|---|
| Up to 0.2 | High terrace | 3614.36 | 0.27 | 366,196.88 | 27.76 |
| | Eluvial with Rock-ridge | 201,399.81 | 15.27 | | |
| | Colluvial | 56,792.05 | 4.31 | | |
| | Deluvial colluvial | 103,281.75 | 7.83 | | |
| | Tukulan (eolian) | 1108.91 | 0.08 | | |
| 0.2–0.4 | Marshes | 18,424.19 | 1.40 | 345,451.31 | 26.19 |
| | Mid terrace | 184,452.36 | 1.40 | | |
| | Deluvial solufluction | 276,738.92 | 20.98 | | |
| | Glacial-valley | 7882.70 | 0.60 | | |
| | Valley-sea terrace | 12,939.76 | 0.98 | | |
| | Outwash | 11,013.39 | 0.83 | | |
| 0.4–0.6 | Low terrace | 211,195.38 | 16.01 | 360,128.62 | 27.30 |
| | Alas | 95,338.82 | 7.23 | | |
| | Moraine | 50,943.82 | 3.86 | | |
| | Inter-ridge-lowland | 2650.60 | 0.20 | | |
| 0.6 or more | Inter-alas | 239,152.71 | 18.13 | 247,433.40 | 18.76 |
| | Non-drained mari | 8121.18 | 0.62 | | |
| | Glacier | 159.51 | 0.01 | | |
| Total | | 1,319,210.21 | 100 | 1,319,210.21 | 100 |

Very ice-rich (0.4 to 0.6) landscapes include alasses, moraines, inter-ridge lowland and low terrace terrain type. The area of very-ice rich terrain comprises approximately 27% of Arctic Yakutia. Alasses are the result of ice-complex degradation during warmer periods of the Upper Pleistocene and Holocene and occupy about 7.2% of the map region. Thermokarst lake and alas formation began during the Pleistocene–Holocene transition, 10–12.5 thousand years BP [62]. The distribution of alasses depends on landscape conditions. In the tundra they prevail, comprising up to 72% of the area [63], while in the northern open woodland zone, alasses are significantly less in extent compared to the ice-complex areas.

The wet climate of the tundra and open woodland zones is responsible for high ground ice contents in the alasses with recent ice wedge development. The upper deposits of alas origin, 3–5 m in thickness, are composed of ice-rich fine-grained material with ice wedges of Holocene age [62]. In the Primorsky Lowland, the alas deposits have ice contents of 0.4 in the lower Indigirka area, 0.5 in the watershed between the Alazeya and Chukochya rivers, and 0.6 in the lower Kolyma basin [31].

In recent decades, thaw-lake drainage has been active similar to that observed in northern North America [64]. Traditional land reclamation activities to create alas meadows by lake drainage proved unsuccessful in the Arctic due to moss and brush overgrowth, as well as due to frost cracking and thermokarst resulting in polygonal ground and patterned bogs [65,66].

Moraines occur in the foothill areas as moraine-ridge complexes. Extensive moraine areas are found in the intermontane depressions. Ice lenses and layers forming cortical, basal and, occasionally, lenticular, and reticulate cryostructures are abundant in the accumulation zone of glacial deposits, while thin ice wedges are present in the glacier ablation

zone. Buried glacier, lake and intrusive ice are likely to be present. Moraines are composed of gravel- to boulder-size material with silt and sand fines, while morainal depressions consist of finer, ice-saturated material.

The inter-ridge lowland terrain type, which occupies vast inter-ridge depressions of mid-high terraces, is always saturated and its deposits are overlain by peat. The dominant cryostructures are massive, lenticular, and layered. Ice wedges are of recent age and 4 to 6 m high.

The low terrace terrain type (about 16% of the map region) comprises the fluvial floodplains and first terraces and covers the large delta areas. Very high ice contents in the low terraces are due to the deposition of alluvium under the humid climatic conditions prevailing today. The sediments dominantly contain massive, lenticular, and layered cryostructures. The low fluvial terraces in the Arctic contain Holocene ice wedges characteristically 4 to 6 m high and 0.5 to 2 m wide, with average polygon sizes of 8 to 12 m.

Ice-rich (0.2 to 0.4) landscapes cover about 26% of Arctic Yakutia. The deluvial-solifluction slope terrain type is most extensive in this ground ice class, covering approximately 21% of the total area. Deluvial-solifluction deposits are characteristic of gentle slopes of the plateaus and mountains and generally consist of fine-grained material. They contain layered, lenticular, massive and reticular cryostructures. Sediments in the footslope areas have higher ice contents. Gentle slopes in the Central Siberian Plateau may contain dells—flat-bottomed depressions on ice-rich silts formed partially due to thermokarst [67]. Drunken' forests with crooked trees occur at the footslopes due to solifluction. Ice-rich slope deposits in Arctic Yakutia were investigated in detail by Gravis [39] who attributed the origin of the ice complex with ice wedges to slope solifluction.

About 4% of the map region are occupied by the mid-terrace, glacial-valley, valley-sea terrace, and outwash terrain types. The mid terrace terrain has been best studied on Arga-Muora-Sise Island in the Lena Delta. Sediments here consist of alluvial sands with massive cryostructure and contain thin polygonal wedge ice [68]. Tabular ground ice bodies occur occasionally. The outwash terrain occurs in the foothill areas of the Verkhoyansk and Chersky Mountains. Sands with pebbles characteristic for this terrain type contain massive, lenticular, and cortical cryostructures. The glacial valley terrain type occurs in the Verkhoyansk and Chersky Mountains and consists of boulders and gravels with sand and silt fines; cryostructures are massive, cortical, and basal. The valley-sea terrace terrain has been studied in the Khallerchinskaya tundra, lower Kolyma area, where dominating sands contain massive cryostructure, while sandy and clayey silts have lenticular cryostructure.

The marsh terrain type occurs on the Laptev and East Siberain seacoasts, as well as on the Arctic islands. The marine sediments are characterized by alternating layers of sand and sandy silt with massive and lenticular cryostructures; total ice contents range between 0.2 and 0.4 [69]. The marine terraces occasionally contain recently developed polygonal ice wedges. The sediments are often overlain by lagoonal lacustrine-palustrine ice-rich deposits.

Ice-poor landscapes occur in about 28% of Arctic Yakutia. The eluvial and rock-ridge terrain types comprising the upland and mountain ridgetops occupy about 15% of the total area of Arctic Yakutia. The eluvial sediments, commonly up to 3–4 m in thickness, are composed of gravel-size rock fragments characterized by cortical cryostructure. Cryostructures in the residual material vary depending on grain-size of the material, being lenticular and layered in silts, massive in sands, and basal in gravels and boulders. Bedrock dominantly contains fissure cryostructure and has very low ice contents (0.05–0.1).

Surficial sediments in the colluvial and deluvial-colluvial slope terrain types are composed of gravel- to boulder-size material and, therefore, the cryostructures are cortical and basal. The fines are characterized by massive cryostructure in sands and lenticular in silts. Rock streams (kurums) occur occasionally on the sloping terrain and infiltration ice accumulates under the kurum layer [70].

The high terrace and tukulan terrain types are uncommon in Arctic Yakutia. Small fragments occur in the border areas which consist of sands with massive cryostructure.

## 4. Discussion

This paper presents the GIS-interpretation (modeling) based on the Permafrost Landscape Map of the Republic of Sakha (Yakutia) [28]. For this, the GIS attribute tables were used as the main mapping method that reflects the relationship between permafrost and landscapes. The first version of the modeling based on the attribute tables of the Permafrost-Landscape Map was synthetic and resulted in cartographic models of ground temperature, ice content of surficial deposits, active layer thickness, and distribution of cryogenic processes [71]. It was a methodological work which presented the capabilities of modeling based on the Permafrost-Landscape Map. In the present work, we have made a regional analysis of the distribution of ground ice volumes in the permafrost landscapes of Arctic Yakutia. Ice content is an important characteristic which controls the response of permafrost to climate change and anthropogenic impacts. Therefore, the compilation of regional maps of permafrost ice content is necessary to understand the dynamics and evolution of permafrost.

There is a relationship between ground ice content and the stratigraphic–genetic types of Quaternary deposits in Arctic Yakutia [72,73]. The ice content map demonstrates regional differentiation of ice content. The compilation principle used for this map is close to the approach adopted for the Circum-Arctic Map of Permafrost and Ground Ice Conditions by Brown et al. [74]. The Brown et al.'s 1:10,000,000-scale map had a purpose of portraying the existing types of permafrost in the Northern Hemisphere and their relative ice contents. Our purpose is to show, based on the global differentiation of permafrost regions by ice content, the structure of ice content of surficial sediments for one large region.

Earlier, Popov et al. [25] compiled the Cryolithological Map of the USSR, at a scale of 1:4,000,000, which showed the genetic types of sediments and their dominant cryostructures. The map was intended for educational purposes and was good enough to display the general distribution of permafrost cryostructures across the USSR. Volumetric ice contents were not shown on the map itself but given for selected cryolithological cores representing the typical permafrost regions of the USSR. Cryostructures and volumetric ice contents were also presented on the Geocryological Map of the USSR, scale 1:2,500,000 [26]. However, this map was based on the genetic sediment types for Quaternary deposits and formation types for pre-Quaternary rocks in the systematization of the ice content of frozen rocks which did not allow equal display of the structure of regions with Quaternary deposits and bedrocks.

The present mapping is comparable in procedures and database to the Circum-Arctic Map of the Yedoma Permafrost Domain [53] which had a purpose of displaying ice-rich areas especially prone to degradation due to climate change or human activity. It is also similar to the map of ground ice in Alaska [75] which based on surficial geology. The map showed ground ice volumes in several classes (>40, 10–40, and <10%), with a primary purpose of delineating ice-rich and ice-poor areas.

In summary, our mapping effort is based on the same principles used in compiling the previous maps, while working to improve those aspects received little attention in the past. First, a single, stratigraphic-genetic, principle was adopted for assessing ice contents both in sediments and rocks. Second, the mapping effort was ecologically oriented, focusing on the top 5–10 m of permafrost most sensitive to global climate warming and anthropogenic impacts. The classification of ground ice contents at 0.2 increment will allow to better rank the susceptibility of permafrost landscapes.

Our research had some uncertainties that we could not overcome. This primarily applies to the zoning of ice-rich areas, which is very important for environmental assessment, such as was done by Shmelev et al. [31]. Also, in future studies on the ice content of permafrost, attention should be paid to the depth of occurrence of wedge ice and the thickness of the protective (intermediate) layer. Of course, the databases on permafrost ice content will expand over time, the methods of studying, modeling and mapping will improve, and this is a natural process, and we should strive for this.

## 5. Conclusions

The ground ice map of Arctic Yakutian permafrost landscapes shows that about 19% of the area is occupied by the inter-alas (yedoma) terrain and poorly drained larch bogs (mari) with ultra ice-rich (>0.6) sediments, and by glaciers. Very high ice volumes (0.4–0.6) are found in four terrain types which cover approximately 27% of Arctic Yakutia, with the low-terrace and alas types being most extensive. Moderate ice volumes (0.2–0.4) are characteristic for 25% of the area, occupied predominantly by the deluvial–solifluction terrain type. Low ice volumes (<0.2) commonly occur in the Central Siberian Plateau, the Verkhoyansk and Chersky Mountains, as well as in the high plateaus of north-eastern Siberia, comprising, in total, about 29% of Arctic Yakutia.

The mapping has allowed us to obtain a general understanding of the ground ice conditions in Arctic Yakutia that can be used for environmental management planning with consideration for climate warming and human impacts. The map can provide a basis for constructing thematic maps needed to address the present and future economic and environmental issues in the arctic regions of the Republic of Sakha (Yakutia).

**Author Contributions:** Conceptualization, A.N.F., M.N.Z. and I.P.S.; methodology, A.N.F., N.F.V. and Y.I.T.; software, Y.I.T.; validation, P.Y.K., A.N.F. and M.N.Z.; investigation, P.Y.K., N.F.V., N.I.B., V.A.A., A.G.S., V.P.S. and A.R.D.; writing—original draft preparation, A.N.F., N.F.V. and P.Y.K.; writing—review and editing, A.N.F., M.N.Z., Y.I.T., A.R.D. and I.P.S.; supervision, M.N.Z. and I.P.S.; project administration, A.N.F.; funding acquisition, M.N.Z. and I.P.S. All authors have read and agreed to the published version of the manuscript.

**Funding:** This study was basically supported by the Russian Ministry of Science and Higher Education (Grant "Priority-2030"to Tomsk State University), and RMSHE (AAAA-A20-120111690009-6 "Cryogenic Processes and Natural Risks Associated with the Development of Permafrost Landscapes in Eastern Siberia"). We also thank RFBR for their partial support (projects: 19-29-05151, 20-55-71005, 21-55-75004).

**Institutional Review Board Statement:** Not applicable.

**Data Availability Statement:** The data presented in this study are available on request from the corresponding author.

**Acknowledgments:** We are grateful to the researchers of Melnikov Permafrost Institute of SB RAS, Yakutsk, Russia, for providing long-term investigation data on Siberian Permafrost, and Tomsk State University, for organizing joint Arctic research. We thank editors and anonymous reviewers for their helpful comments that improved the manuscript.

**Conflicts of Interest:** The authors declare no conflict of interest.

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
