# Peer review of "Ice Volumes in Permafrost Landscapes of Arctic Yakutia"

_land, doi:10.3390/land11122329_

Round 1

Reviewer 1 Report

The ground ice and its melting in Yakutia are very important in the regional and globally. This paper present succinctly on the features of ground ice in the key zone of Russia and northern hemisphere. The language and technicality are sound and I believe that it is a great paper. Thus, I recommend accept with minor corrects in some spellings. I also provide with a marked MS. However, it is only a quick look through. It is up to the authors to painstakingly make the paper presented in the best possible form.

Author Response

Thank you very much for viewing the manuscript and your rating. The remarks made in MS all are corrected.

Reviewer 2 Report

General notes

Specify for what time slice this map was made, whether it is permissible to use data from old maps of 1985, 1985, 1996, 2009 editions. How quickly does the landscape change over time, according to the authors?

Add to bibliography DOI

Line 16-18 it is need to remove the goal of study from  Abstract text .  It is repeated in Introduction (Line 67-69).

Line 31 add that this is the air temperature

Line 58-63  Information on cryostructures, wedge ice and volumetric ice contents is presented in  the permafrost maps for the USSR (Russia) published previously, including the Cryolith-59 ological Map of the USSR, 1:4,000,000 scale [25] and the Geocryological Map of the CIS, 1:2,500,000 scale [26]. The Permafrost-Landscape Maps of Yakutia [27-28] also contain information on ground ice conditions. Smaller-scale maps are available for various regions 2 of Yakutia showing the cryostratigraphic characteristics of permafrost [24, 29-31]. Have these maps always characterized the upper 5-10 m of permafrost? Can this information be used?

Line 107-119 Data on ground ice volumes for the compilation of the above maps were drawn both  from published [31, 37-46] and unpublished sources available at the Melnikov Permafrost  Institute. The unpublished sources included the reports of the First Northern Expedition  led by N.F. Grigoriev (1954–1957) and the Second Northern Expedition led by Y.V. Shumilov (1984–1987), as well as field records by E.M. Katasonov, I.A. Nekrasov, P.P. Gavriliev, I.V. Klimovsky, V.V. Kunitsky and many other researchers. Have these maps always characterized the upper 5-10 m of permafrost? Can this information be used?

Line 122   what polygons are we talking about? Polygon size?

Line 134 Is the information on the map «Ground ice content map of Arctic Yakutia»  different from the published «map of ground ice content (in fraction)» figure 5. (Shestakova, A.A.; Fedorov, A.N.; Torgovkin, Y.I.; Konstantinov, P.Y.; Vasiliev, N.F.; Kalinicheva, S.V.; Samsonova, V.V.; 492 Hiyama, T.; Iijima, Y.; Park, H.; Iwahana, G.; and Gorokhov, A.N. Mapping the main characteristics of permafrost on the basis of a permafrost-landscape map of Yakutia using GIS. Land 2021, 10, 5, 462.) ? What is the novelty?

Geographical names are hard to read on the map. It is necessary to show all the names mentioned in the text on the map, at least in numbers. How are glaciers shown? Why are they characterized by ice content?

Why there are Glacier ice in "Ground ice content map of Arctic Yakutia" as a Terrain type ?

Line 136-137 only 18 Terrain type? Where are 19? You need to decrypt the stratigraphic-genetic complex indexes in table 1.

Line 174-177   The distribution of alasses depends on landscape conditions. In the tundra they prevail, comprising up to 72% of the area [63], while in the northern open woodland zone, alasses are significantly less in extent compared to the ice-complex areas. How do the authors explain this pattern?

Line 216 Valley-sea terrace = Laida?

Line 250   Attach supplementary or link to GIS attribute tables

Line 263-264 The ice content map demonstrates  regional differentiation of ice content depending on the permafrost-landscape structure. This output is not readable on the ice content map.

Line 266 The Brown  et al.’s 1:10,000,000-scale map had a purpose of portraying the existing types of permafrost  in the Northern Hemisphere and their relative ice contents. But, Brown's map only shows permafrost distribution, notice content.

Author Response

Many thanks for the review of the manuscript and for the comments.

Response to Reviewer 2 Comments

Point 1: Specify for what time slice this map was made, whether it is permissible to use data from old maps of 1985, 1985, 1996, 2009 editions. How quickly does the landscape change over time, according to the authors?

Response 1: The map was compiled for a modern time. Since the map was compiled on the basis of geological and geomorphological classifications - stratigraphic-genetic complexes (SGC), the change in the content of SGC is subject to the geological evolution. Some changes may occur in local areas, which can be displayed on plans or detail scale maps, but not on a scale of our map. Therefore, the time scales of 1985, 1985, 1996, 2009 for compiling the "Ground ice content map of Arctic Yakutia" do not play a role.

Point 2: Add to bibliography DOI.

Response 2: The LAND template DOI was not recommended, so we didn't include it in the reference.

Point 3: Line 16-18 it is need to remove the goal of study from  Abstract text.  It is repeated in Introduction (Line 67-69).

Response 3: Corrected in the text of the Abstract: This article is devoted to the study of the distribution of ground ice volumes in the upper layers of 5-10 m permafrost in the permafrost landscapes of Arctic Yakutia. Compilation of such a map will serve as a basis for assessing the vulnerability of permafrost to global warming, anthropogenic impact and forecasting the evolution of permafrost landscapes.

Point 4: Line 31 add that this is the air temperature.

Response 4: Added: For the Russian Arctic, a 7°C increase in air temperature is predicted by the end of the 21st century relative to the late 20th century [1].

Point 5: Line 58-63  Information on cryostructures, wedge ice and volumetric ice contents is presented in  the permafrost maps for the USSR (Russia) published previously, including the Cryolithological Map of the USSR, 1:4,000,000 scale [25] and the Geocryological Map of the CIS, 1:2,500,000 scale [26]. The Permafrost-Landscape Maps of Yakutia [27-28] also contain information on ground ice conditions. Smaller-scale maps are available for various regions 2 of Yakutia showing the cryostratigraphic characteristics of permafrost [24, 29-31]. Have these maps always characterized the upper 5-10 m of permafrost? Can this information be used?

Response 5: On some maps, monographs and scientific articles, the volumetric ice content is given even deeper. For example, in the Cryolithological Map of the USSR, 1:4,000,000 scale [25], the volumetric ice content in cryolithological applications is given up to several hundred meters. However, there is not enough such data for compiling a map, so we chose data only for the upper 5–10 m of permafrost. Many cartographic works give the total thickness of the stratigraphic-genetic complex, while methodological articles and monographs stipulate that they take the upper parts of the permafrost.

Point 6: Line 107-119 Data on ground ice volumes for the compilation of the above maps were drawn both  from published [31, 37-46] and unpublished sources available at the Melnikov Permafrost  Institute. The unpublished sources included the reports of the First Northern Expedition  led by N.F. Grigoriev (1954–1957) and the Second Northern Expedition led by Y.V. Shumilov (1984–1987), as well as field records by E.M. Katasonov, I.A. Nekrasov, P.P. Gavriliev, I.V. Klimovsky, V.V. Kunitsky and many other researchers. Have these maps always characterized the upper 5-10 m of permafrost? Can this information be used?

Response 6: Usually, in geocryological studies, ice content is indicated by lithological differences and depths, so we chose only the upper 5-10 m as the most vulnerable parts of the permafrost in global warming and anthropogenic disturbance. The unpublished data from the Melnikov Permafrost Institute are numerous enough to be used for mapping purposes.

Point 7: Line 122   what polygons are we talking about? Polygon size?

Response 7: Here we are talking about cartographic GIS polygons. Therefore, the size of the polygons is inappropriate.

Point 8: Line 134 Is the information on the map «Ground ice content map of Arctic Yakutia»  different from the published «map of ground ice content (in fraction)» figure 5. (Shestakova, A.A.; Fedorov, A.N.; Torgovkin, Y.I.; Konstantinov, P.Y.; Vasiliev, N.F.; Kalinicheva, S.V.; Samsonova, V.V.; 492 Hiyama, T.; Iijima, Y.; Park, H.; Iwahana, G.; and Gorokhov, A.N. Mapping the main characteristics of permafrost on the basis of a permafrost-landscape map of Yakutia using GIS. Land 2021, 10, 5, 462.) ? What is the novelty?

Response 8: “Ground ice content map of Arctic Yakutia” is a regional map that reflects specific data on the ice content of permafrost landscapes in Arctic Yakutia, these data differ from data from other regions of Yakutia. The article by Shestakova et al. [71] devoted to the methodological issues of GIS modeling, where the average characteristics of ice content throughout Yakutia were conventionally taken.

Point 9: Geographical names are hard to read on the map. It is necessary to show all the names mentioned in the text on the map, at least in numbers. How are glaciers shown? Why are they characterized by ice content?

Response 9: Comment accepted. However, it is very difficult to display maps for large regions in the article, so we decided to give the web address of the ice cover map and the permafrost-landscape map [28] in Supplementary Materials: https://cloud.mail.ru/public/8j62/vJpw5hEGa/Ground_ice_content_of_Arctic_Yakutia.pdf and https://cloud.mail.ru/public/8j62/vJpw5hEGa/mlk_2020.pdf (The Permafrost-Landscape Map of Sakha (Yakutia) Republic [28]).

Point 10: Why there are Glacier ice in "Ground ice content map of Arctic Yakutia" as a Terrain type?

Response 10: Glaciers as terrain types represent a landscape category [28], no glaciers are indicated on the “Ground ice content map of Arctic Yakutia”, the ice content is “more than 0.6” in Table 1.

Point 11: Line 136-137 only 18 Terrain type? Where are 19? You need to decrypt the stratigraphic-genetic complex indexes in table 1.

Response 11: Referring to the previously compiled base “The Permafrost-Landscape Map of Sakha (Yakutia) Republic” [28], we distinguish 19 types of terrain. However, when compiling the “Ground ice content map of Arctic Yakutia”, we combined eluvial and rock types of terrain (see table 1, unit 10 and note).

Point 12: Line 174-177   The distribution of alasses depends on landscape conditions. In the tundra they prevail, comprising up to 72% of the area [63], while in the northern open woodland zone, alasses are significantly less in extent compared to the ice-complex areas. How do the authors explain this pattern?

Response 12: It really depends on the landscape conditions (humidity, forest cover), in waterlogged tundra alases occupy large areas, but in the relatively dry middle taiga of Central Yakutia - only up to 19% (Bosikov, 1978).

Point 13: Line 216 Valley-sea terrace = Laida?

Response 13: The valley-sea terrace is formed by complex alluvial-marine deposits, they are not associated with sea tides, and lays are developed on marine deposits, associated with sea tides, they are part of the marsh type of terrain.

Point 14: Line 250   Attach supplementary or link to GIS attribute tables

Response 14: The next sentence contains a link to Shestakova et al. [71] for a description of modeling from a GIS attribute table.

Point 15: Line 263-264 The ice content map demonstrates  regional differentiation of ice content depending on the permafrost-landscape structure. This output is not readable on the ice content map.

Response 15: Agree. Text modified: The ice cover map shows the regional differentiation of ice content.

Point 16: Line 266 The Brown  et al.’s 1:10,000,000-scale map had a purpose of portraying the existing types of permafrost  in the Northern Hemisphere and their relative ice contents. But, Brown's map only shows permafrost distribution, notice content.

Response 16: In the map legend of Brown et al., the ice content is given:

https://data.tpdc.ac.cn/en/data/c66bf4a7-8f20-443c-9412-53ac675bd964/
